# Oleanolic Acid Derivative AXX-18 Exerts Antiviral Activity by Inhibiting the Expression of HSV-1 Viral Genes UL8 and UL52

**DOI:** 10.3390/v14061287

**Published:** 2022-06-13

**Authors:** Zhaoyang Wang, Jiaoyan Jia, Yuzhou Jiang, Feng Li, Yiliang Wang, Xiaowei Song, Shurong Qin, Yifei Wang, Kai Zheng, Binyuan Hu, Yongxian Cheng, Zhe Ren

**Affiliations:** 1Department of Cell Biology, College of Life Science and Technology, Jinan University, Guangzhou 510632, China; WZY110906@163.com (Z.W.); jjy931017@163.com (J.J.); 13168697011@163.com (Y.J.); li_fengfeng@163.com (F.L.); yiliang_wang@foxmail.com (Y.W.); xiaowei@stu2017.jnu.edu.cn (X.S.); DG20340064@NJU.EDU.cn (S.Q.); twang-yf@163.com (Y.W.); 2Guangdong Province Key Laboratory of Bioengineering Medicine, Guangzhou 510632, China; 3Guangdong Provincial Biotechnology Drug and Engineering Technology Research Center, Guangzhou 510632, China; 4National Engineering Research Center of Genetic Medicine, Guangzhou 510632, China; 5The Key Laboratory of Virology of Guangdong, Guangzhou 510632, China; 6Institute for Inheritance-Based Innovation of Chinese Medicine, School of Pharmaceutical Sciences, Health Science Center, Shenzhen University, Shenzhen 518060, China; zhengk@szu.edu.cn (K.Z.); hubinyuan2018@163.com (B.H.); 7School of Life Science and Food Engineering, Hanshan Normal University, Chaozhou 521041, China

**Keywords:** herpes simplex virus type 1, oleanolic-acid derivative AXX-18, antiviral activity, helicase-primer enzyme complex, UL8, UL52

## Abstract

Two-thirds of the world’s population is infected with HSV-1, which is closely associated with many diseases, such as Gingival stomatitis and viral encephalitis. However, the drugs that are currently clinically effective in treating HSV-1 are Acyclovir (ACV), Ganciclovir, and Valacyclovir. Due to the widespread use of ACV, the number of drug-resistant strains of ACV is increasing, so searching for new anti-HSV-1 drugs is urgent. The oleanolic-acid derivative AXX-18 showed a CC_50_ value of 44.69 μM for toxicity to HaCaT cells and an EC_50_ value of 1.47 μM for anti-HSV-1/F. In addition, AXX-18 showed significant inhibition of ACV-resistant strains 153, 106, and Blue, and the anti-HSV-1 activity of AXX-18 was higher than that of oleanolic acid. The mechanism of action of AXX-18 was found to be similar to that of oleanolic acid, except that AXX-18 could act on both the UL8 and UL52 proteins of the uncoupling helicase-primase enzyme, whereas oleanolic acid could only act on the UL8 protein. We have elucidated the antiviral mechanism of AXX-18 in detail and, finally, found that AXX-18 significantly inhibited the formation of skin herpes. In conclusion, we have explored the anti-HSV-1 activity of AXX-18 in vitro and in vivo as well as identification of its potential target proteins, which will provide a theoretical basis for the development of subsequent anti-HSV-1 drugs.

## 1. Introduction

Herpes simplex virus (herpes simplex virus, HSV) is a double-stranded DNA virus, including herpes simplex virus type 1 (HSV-1) and herpes simplex virus type 2 (HSV-2) [1,2]. The prevalence of HSV-1 in humans is very high, as two-thirds of the global population has been infected with HSV-1 [3,4,5]. HSV-1 mainly infects the mouth, lips, and face of humans, and can be latent in the trigeminal ganglion for a long time. HSV-1 infection is related to many diseases, such as Gingival stomatitis and viral encephalitis [6], the most common of which is viral encephalitis [7]. In addition, it has been shown that reactivation of HSV-1 is associated with a strong risk of death and pneumonia in patients with severe COVID-19 [8]. HSV-1 proliferates by infecting host cells, and its proliferation process mainly includes adsorption, puncture, and nuclear entry, as well as viral-genome replication, assembly, and release stages [9]. The replication stage of the HSV-1 genome occurs in the nucleus. The process mainly includes three stages: immediate-early (IE), early (E), and late (L) [10]. In addition, the transcriptional activation of E and L genes requires the regulation of IE-gene products [11]. In other words, the genome replication of HSV-1 is a cascade effect. Theoretically, if the drug can significantly inhibit the expression of the immediate-early gene (IE) in the immediate-early stage, then early (E) and late (L) HSV-1 gene expression will also be inhibited, and ultimately inhibit the proliferation of HSV-1.

Seven viral proteins are required in the early stages of viral-genome replication, including the helicase-primer enzyme complex (UL5-UL8-UL52), DNA polymerase-synthesis enzyme complex (UL30-UL42), single-stranded DNA-binding protein (UL29), and origin-binding protein (UL9) [12,13]. HSV-1 helicase-primer enzyme is a complex composed of UL5, UL8, and UL52, three viral-gene-coding products. Its function is to untie the viral DNA double helix and synthesize single-stranded DNA primers to initiate DNA replication [14]. The other complex is a polymerase-synthetase complex, composed of UL30-U42, with a main function of synthesizing double-stranded DNA [15], and ACV, which exerts antiviral activity by targeting this complex [16]. ACV is an anti-HSV clinical drug but due to the gradual emergence of ACV-resistant strains because of the wide application of ACV, developing new anti-HSV drugs and exploring new anti-HSV targets is urgent. Currently, several drugs have been developed for this complex, including BILS179BS, AIC316 (formerly known as BAY 57-1293), and ASP2151 (amenamevir), some of which have been used in phase II clinical applications, and their anti-HSV-1 mechanism is different from that of ACV [17]. In addition, as these complexes have no homologs in eukaryotes, and they are essential for HSV-1 genome replication, helicase-chimeras complexes are worthy targets for development against HSV-1 [18].

With the improvement of natural medicine-separation technology and the advancement of anti-HSV-1 screening methods, the role and position of natural products in the research of new anti-HSV drugs has become increasingly prominent. Therefore, we are looking for high-efficiency, low-toxicity, anti-HSV active ingredients from natural medicines. This is a shortcut to obtain lead compounds and is of great significance to the research and development of new anti-HSV drugs with independent intellectual property rights.

In this study, AXX-18 is a derivative of oleanolic acid, which is separated and extracted from benzoin. In our previous study, we reported on the anti-HSV-1 activity and mechanism of oleanolic acid as well as elucidated the target protein of oleanolic-acid action [19]. In addition, our preliminary study found that the anti-HSV-1 activity of the oleanolic-acid derivative AXX-18 was superior to that of oleanolic acid, so we conducted an in-depth study on the antiviral activity and mechanism of AXX-18.

## 2. Materials and Methods

### 2.1. Cells and Viruses

African green monkey kidney cell line (Vero cells) and Human immortalized keratinocyte cell line (HaCaT) were purchased from the American Type Culture Collection Center (ATCC) and cultured in Dulbecco’s Modified Eagle Medium (DMEM; 8118305, GIBCO/Thermo Fisher Scientific, Waltham, MA, USA) with 10% fetal bovine serum (FND500, Excell Bio, Shanghai, China). SH-SY5Y cells (ATCC CRL-2266) were cultured at 37 °C in a humid atmosphere with 5% CO_2_. HSV-1/Blue, a TK mutant derived from HSV-1 (KOS), and two acyclovir-resistant clinical HSV-1 strains, HSV-1/106 and HSV-1/153, were kind gifts from Tao Peng [20], State Key Laboratory of Respiratory Disease, Guangzhou Institutes of Biomedicine and Health, Chinese Academy of Sciences. GFP-HSV-1, expressing a GFP-tagged viral protein Us11, was used to evaluate viral-nuclear transport; HSV-1 strain F (ATCC, Manassas, VA, USA), initially obtained from Hong Kong University, was propagated in Vero cells and stored at −80 °C.

### 2.2. Compounds, Antibodies, siRNAs, and Plasmids

Oleanolic-acid derivative AXX-18 is isolated from benzoin, and its separation process is completed by Shenzhen University. AXX-18 was dissolved in Dimethyl Sulphoxide with a concentration of 50 mM. The gel composition of the AXX-18 gel formulation is composed of 99% glycerol (G10007) and 1% carbomer U20 (M64011). Acyclovir (ACV) was purchased from Sigma-Aldrich (St. Louis, MO, USA). Anti-VP16 (ab11026), Anti-gB (ab6506), Anti-ICP0 (ab6513), Anti-ICP4 (ab6514), Anti-ICP8 (ab20194), Anti-ICP27 (ab53480), Anti-ICP22 (ab6506), Anti-gD (ab6507), Anti-FLAG (ab205606), Anti-HA (ab236632), Anti-GAPDH (ab9485), and Anti-β actin (ab9485) antibodies were purchased from Abcam (Cambridge, UK). All the eukaryotic expression plasmids, including p3Xflag-UL5, pCMV-HA-UL8, and pEGFPC1-UL52, were generated in our laboratory. All plasmids and primer information are shown in Appendix A. All siRNAs sequences were purchased from Gene Pharma (Shanghai, China), and the information is shown in Appendix A. All qRT-PCR primer information is shown in Appendix A.

### 2.3. Cytotoxicity and Antiviral Activity Assay

Cultured Vero cells were placed into a 96-well plate, 1 × 10^4^ cells/well; the next day, different concentrations of AXX-18, OC, DMSO, or ACV were added, and incubation was continued for 72 h, then 5 μL/well of CCK8 were added and incubated at 37 °C for 1 h, and, finally, the OD value was assessed by an enzyme immunoassay reader at 480 nm.

A plaque reduction test to explore drug antiviral activity was completed, and Vero cells were inoculated into a 24-well plate, 2 × 10^5^ cells/well; the next day, HSV-1 was infected with cells at 100 PFU/well, adsorbed at 37 °C for 2 h, and, after that, it was changed to the covering solution containing AXX-18 (20 μM), where it continued to cultivate for 72 h, finally being stained with crystal violet, and the number of plaques was counted.

The copy number of viral genome DNA to explore the antiviral activity of drugs was detected andHaCaT cells were inoculated into a 24-well plate; the next day, HSV-1 was infected with cells at MOI = 1 and was adsorbed at 37 °C for 2 h, then AXX-18 (20 μM) or OC (20 μM) was added, after being cultured for 24 h, the supernatant and cells were collected, they were placed at −80 °C and were frozen and thawed three times, then viral genomic DNA was extracted, and the copy number of viral genomic DNA was detected by qRT-PCR.

### 2.4. Quantitative Real-Time PCR (qRT-PCR)

Total RNA from cells with TRIzol reagent was extracted, (TIANGEN, Beijing, China) in accordance with the protocol of the manufacturer, the RNA concentration was measured at 260/280 nm using a NanoPhotometer P330 spectrophotometer (IMPLEN, Munich, Germany), and 1 μg RNA was then reverse transcribed into cDNA using a PrimeScript RT Reagent Kit (TAKARA, Dalian, China). Subsequently, the mRNA expression levels of viral genes were analyzed using a Bio-Rad CFX96 real-time PCR system (Bio-Rad, Hercules, CA, USA), according to previous studies, and *GAPDH* was used as a reference in HaCaT cells.

### 2.5. Virus Inactivation Assay

HaCaT cells were cultured into 24-well plates, 2 × 10^5^ cells/well; the next day, AXX-18 and HSV-1 were incubated together for 2 h at 37 °C, then the cells were infected with this mixture for 2 h, then the cover was changed and they were continued to be incubated for 72 h, finally, crystal violet staining was performed and the number of plaques was counted to calculate the plaque inhibition rate.

### 2.6. Virus Attachment Assay

HaCaT cells were cultured into 24-well plates, 2 × 10^5^ cells/well; the next day, cells were pre-cooled at 4 °C for 1 h and washed with cold PBS. Virus inoculum (100 PFU/well) and AXX-18 (20 μM) were added at the indicated concentrations into the cell wells, and the mixture was incubated at 4 °C for another 2 h to allow the virus to attach to the cells. The virus inoculum was removed. Cells were replenished with a cover layer, and 72 h later, they were fixed and stained as described above.

### 2.7. Virus Penetration Assay

HaCaT cells were cultured into 24-well plates, 1.5 × 10^5^ cells/well; the next day, cells were pre-cooled at 4 °C for 1 h, subsequently, HSV-1 was infected with the cells for 2 h at 4 °C, then HSV-1 was removed and added to AXX-18 (20 μM) to be incubated at 37 °C for 10 min. After incubation, PBS (pH = 3) was added to every well for 1 min to inactivate the virus, which failed to penetrate the cells. After that, the solution was neutralized, and the neutral PBS was removed. Cells were replenished with a cover layer and 72 h later, they were fixed and stained as described above.

### 2.8. Virus into the Nuclear Assay

HaCaT cells were cultured to a confocal dish, 1 × 10^5^ cells/well; the next day, HaCaT cells were infected with HSV-1 (MOI = 1) or HSV-1 and AXX-18 (20 μM) mixed solution for 2 h at 37 °C; HSV-1 was absorbed, discarded, and washed with PBS three times, then closed, permeated, and incubated with primary antibody, fluorescent secondary antibody, and DAPI; and, finally, photographed under fluorescence laser confocal.

### 2.9. Western Blot Assay

Cells were lysed with sodium dodecyl sulfate (SDS) buffer (Beyotime, Shanghai, China) containing 1 M phenyl methyl sulfonyl fluoride (PMSF), and proteins were separated by 8–15% gradient SDS-PAGE, transferred to polyvinylidene fluoride (PVDF) membrane (Millipore), and then blocked with 5% nonfat milk for 1 h at room temperature. The target protein was incubated with the primary antibody overnight at 4 °C, and then the secondary antibody was incubated at room temperature for 1 h. Finally, those target proteins were detected by ECL solutions. The band intensity of each protein was calculated using Quantity One software (Bio-Rad, Hercules, CA, USA) and was normalized to that of βactin.

### 2.10. Transfection of siRNA and Detection of Virus Titer

HaCaT cells were cultured to 12-well plates, 1.5 × 10^5^ cells/well; the next day, cells were infected with HSV-1 (MOI = 1) for 4 h, and then siRNA (100 nM/well) was transfected into HaCaT cells by Lipofectamine 3000 transfection reagent; after 4 h, the contents were aspirated, the PBS was washed, and then the culture was continued at 37 °C for 24 h. Finally, the sample was frozen and thawed three times and the virus titer was checked.

### 2.11. Transfection of Overexpression Plasmid and Detection of Virus Titer

HaCaT cells was cultured to 12-well plates, 1.5 × 10^5^ cells/well, and p3Xflag-UL5, pCMV-HA-UL8, and pEGFPC1-UL52 were transfected into HaCaT cells with Lipofectamine 3000 transfection reagent; after 4 h, the maintenance solution was changed to one containing HSV-1 (MOI = 1) or HSV-1 and AXX-18 (20 μM), and the culture was continued at 37 °C for 24 h. Finally, the sample was frozen and thawed three times and the virus titer was checked.

### 2.12. Luciferase Reporter Gene Assay

The effect of AXX-18 on the promoter activity of viral immediate-early genes was analyzed using a dual luciferase assay. Briefly, the promoter sequence of the viral immediate-early gene α0 and α4 was cloned into the luciferase-reporter plasmid pGL4.12 [luc2p] (Promega, Madison, WI, USA) in accordance with the instructions of the manufacturer. HSV-1 virion protein 16 (VP16) is a crucial protein involved in the assembly of a transactivation complex on the promoters of viral α0 and α4. Therefore, the exogenous expression of VP16 was used as a positive control, and the protein-coding sequence of VP16 was cloned into the expression plasmid pcDNA3.1 (pcDNA). HaCaT cells were transfected with pcDNA3.1(+)-VP16 plasmid (pcDNA-VP16) (250 ng/well) in combination with pGL4.12 [luc2p]-α0 promoter (p-GLα0) plasmid (250 ng/well) or pGL4.12 [luc2p]-α4 promoter (p-GLα4) plasmid (250 ng/well) using a jetPRIME^®^ kit (PT-114-15; Polyplus Transfection, France). The pRL-TK plasmid (5 ng/well) was transfected as an internal reference. After transfection, the cells were treated with AXX-18 (20 µM) for 2 h. A Dual-Luciferase^®^ Reporter assay was performed using a GloMax 20/20 instrument (Promega, Madison, WI, USA).

### 2.13. Animal Experiment

The method of this experiment was based on Yiliang Wang [21]. Briefly, 6–8-week-old C57 BL/6 female mice (Guangdong Medical Laboratory Animal Center) were randomly divided into 4 groups of 7 mice each. The mice were debrided on the back and injected with sodium phenobarbital the next day to lightly paralyze the mice; a 20 μL drop of HSV-1 containing 10^5^ PFU was applied to the skin near the top of the spleen; then, the virus suspension was allowed to dry by scratching the skin 20 times through the drop with a 27-gauge needle. Subsequently, 0.05% AXX-18, 0.1% AXX-18, and ACV gel preparations were used once daily in the mice, and the HSV-1 group was treated with blank gel. Mice were euthanized after 9 days of continuous administration and, subsequently, skin herpes tissue, heart, liver, spleen, lung, and kidney tissue were taken, and the width of the skin herpes was measured. RNA was extracted from the skin tissues using Trizol, and the expression of UL5, UL8, and UL52 mRNA was measured by qRT-PCR. Protein was also extracted from each tissue using SDS, and the expression of the viral protein gD was measured by Western blot. All instruments utilized in obtaining the tissue specimens were washed in methylated spirits between each manipulation. All animal experiments were performed with the approval of the Guangzhou (Jinan) Biomedical Research and Development Center.

### 2.14. Statistical Analysis

Data are presented as mean ± SD. Data were analyzed by one-way analysis of variance or Student’s t-test as appropriate, and the level of significance was set at *p* < 0.05 (*), *p* < 0.01 (**), or *p* < 0.001 (***).

## 3. Results

Figure 1. The anti-HSV-1 activity of oleanolic-acid derivative AXX-18 is significantly higher than that of oleanolic acid.

The toxicity of AXX-18, OC, and ACV on Vero and HaCaT cells was detected by CCK8, and the CC_50_ values are shown in Appendix A. We found that the toxicity of AXX-18 is lower than OC at the same concentration (Figure 1B,C). Subsequently, we compared the anti-HSV-1 activity of AXX-18 and OC. The results of plaque reduction showed that the EC_50_ value of AXX-18 was lower than OC (Figure 1D), which suggested that the antiviral activity of AXX-18 is stronger than that of OC. In addition, the qRT-PCR results show that AXX-18 can inhibit the replication of both intracellular and extracellular viral genomes, and the inhibitory effect of AXX-18 is more significant than that of OC (Figure 1E,F). Subsequently, we also found that AXX-18 treatment of cells for either 12 h or 24 h significantly inhibited virus titers (Figure 1G) and that AXX-18 inhibited significantly more than OC. Therefore, we believe that the anti-HSV-1 activity of AXX-18 is stronger than that of OC.

Figure 2. Against ACV resistant virus strains activity of AXX-18 and OC.

Acyclovir (ACV) exerts antiviral activity, depending on whether the thymine kinase (TK) [22] and clinical isolates of ACV-resistant HSV were TK-negative, TK-low mutant, or TK-altered mutant [23]. We investigated the influence of AXX-18 on ACV-resistant strains. In this study, we used HSV-1/Blue, a TK mutant derived from HSV-1, and two clinical HSV-1/106 and HSV-1/153 [24]. Plaque experiments showed that the viral inhibition rate of AXX-18 to HSV-1/106, HSV-1/153, and HSV-1/Blue at 20 μM was about 100%. Its EC_50_ values were 6.04 μM, 9.29 μM, and 6.78 μM, respectively, and the EC_50_ values of oleanolic acid were 12.74 μM, 15.21 μM, and 12.96 μM (for detailed values see Appendix A), which were significantly higher than those of AXX-18, suggesting that AXX-18 has anti-ACV-resistant strains, and the activity is significantly stronger than that of OC (Figure 2A–C). In order to better explore the point, we also tested on HaCaT cells and found that AXX-18 significantly inhibited the DNA copy number of the immediate-early gene α0 and the late gene UL47 of the ACV-resistant strains 106 and Blue (Figure 2D–F). In addition, the AXX-18 treatment of HaCaT cells resulted in a significant reduction in the titer of the 106 and Blue virus strains (Figure 2E–G). This indicates that AXX-18 also has good activity against ACV-resistant virus strains on HaCaT cells and is stronger than oleanolic acid. On the other hand, it indicates that the anti-HSV-1 mechanism of action for AXX-18 is different from that of ACV, which also provides a theoretical basis for our subsequent mechanistic studies.

Figure 3. AXX-18 acts in the immediate-early stage.

To explore the stage at which AXX-18 exerts its antiviral activity, we used plaque reduction experiments to add drugs at different time points. The results show that added AXX-18 at 0–24 h can significantly reduce the number and size of plaques, but it has higher antiviral activity in the immediate-early stage (0–4 h) (Figure 3B–D), indicating that AXX-18 may act on the immediate-early stage. In order to prove this more fully, we added drugs at different time points on HaCaT cells. The results showed that AXX-18 added at 0–24 h can significantly reduce the virus genome-copy number and virus titer (Figure 3E,F), and AXX-18 added at 0–4 h has the best antiviral effect. In summary, the results suggest that AXX-18 acts as an antiviral effect in the immediate-early stage after HSV-1 infection.

Figure 4. AXX-18 does not affect the attachment, penetration, and nuclear entry of HSV-1.

The immediate-early stages of HSV-1 infection in the host include attachment, penetration, and nuclear entry, so we examined the effect of AXX-18 on these three stages. In order to exclude the effect of AXX-18 on the direct inactivation of HSV-1, we incubated AXX-18 with HSV-1 for 2 h and then infected HaCaT cells. The results showed that AXX-18 did not affect the number of plaques (Figure 4A), suggesting that AXX-18 has no direct inactivation effect on HSV-1. In addition, we also examined the effect of AXX-18 on HSV-1 attachment and penetration by means of a plaque subtraction assay, and the results show that AXX-18 makes no significant difference on the number of plaques (Figure 4B,C). It is suggested that AXX-18 exerts antiviral activity by not affecting the adsorption and puncture of HSV-1. Finally, we performed immunofluorescence staining for the viral protein gB and, thus, observed the effect of AXX-18 on the intracellular localization of HSV-1. The results showed that AXX-18 did not affect the intracellular localization of gB (Figure 4D), indicating that AXX-18 also exerts its antiviral effect by not affecting the entry of HSV-1 into the nucleus.

Figure 5. AXX-18 does not affect immediate-early gene and protein expression of HSV-1.

HSV-1 can enter two modes of infection after infecting host cells, including lytic infection and latent infection. In the lytic infection, the capsid protein VP16 can activate the expression of the α gene and then cause the expression of the β gene and γ gene in a cascade-like manner. Therefore, we speculate that AXX-18 may exert antiviral activity by suppressing the expression of immediately-early genes, but, inconsistent with our guess, the immediate-early gene and protein expression were not affected in the presence of AXX-18 (Figure 5A–F). HSV-1 capsid protein VP16 promotes the formation of a transactivation complex, which binds to the promoters of immediate-early genes to initiate their gene expression [25]. Then, we co-expressed VP16 (pcDNA3.1-VP16) and α0 (pGL-α0) or α4 (pGL-α4) in HaCaT cells for 48 h and showed that AXX-18 did not affect the fluorescence activity induced by VP16-mediated (Figure 5G,H). This suggests that AXX-18 does not exert anti-HSV-1 activity by affecting the expression of immediate-early genes.

Figure 6. UL8 and UL52 is a potential target for AXX-18 to exert antiviral effect.

The previous experiments showed that AXX-18 could significantly inhibit the titer of HSV-1 during the period of 0–24 h. In addition, we also found that AXX-18 did not affect the expression of immediate-early genes and proteins, nor did it affect the attachment, penetration, and unclear entry stage. Therefore, we examined the effect of AXX-18 on HSV-1 replication, and qRT-PCR results showed that AXX-18 significantly inhibited the expression of the helicase-primer enzyme complex gene (UL5, UL8, and UL52) in the immediate-early stage, and UL8 decreased most significantly (Figure 6A–C). Subsequently, we designed related siRNAs of UL5, UL8, and UL52 and verified that the knock down efficiency was above 60% (Figure 6D); NC is the negative control. In addition, we used HSV-1 with an EGFP tag (EGFP-HSV-1), and results showed no significant change in EGFP fluorescence intensity when UL5 alone was knocked down, while EGFP fluorescence intensity was significantly reduced when UL8 or UL52 alone was knocked down (Figure 6E,F), suggesting that the knock down of UL8 or UL52 inhibits HSV-1 replication; we speculated that AXX-18 may also exert antiviral activity through UL8 or UL52. Therefore, we performed knock down and overexpression validation, and results showed that AXX-18 significantly reduced viral titers in the NC group, that AXX-18 did not significantly reduce viral titers when UL8 or UL52 was knocked down (Figure 6G), and that overexpression experiments showed that the reduction in viral titers caused by AXX-18 was significantly restored when UL8 or UL52 was overexpressed (Figure 6H). On the other hand, Western blot assays showed that overexpression of UL8 or UL52 restored the reduction in viral protein gD expression caused by AXX-18, while overexpression of UL5 did not restore viral titer or viral protein (Figure 6I). All the above experiments suggest that AXX-18 exerts its antiviral activity through UL8 and UL52. It also indicates that UL8 and UL52 may be potential target proteins of AXX-18.

Figure 7. AXX-18 can significantly improve skin herpes caused by HSV-1 infection.

In order to investigate the broad spectrum of the anti-HSV-1 activity of AXX-18, it is particularly important to study the anti-viral activity of AXX-18 in mice. Therefore, we constructed a model of skin herpes caused by HSV-1. To facilitate drug delivery, AXX-18 was made into a gel formulation. We infected the skin of mice immediately after trauma with 10^5^ PFU of HSV-1, after which different concentrations of AXX-18 gel and ACV were applied to the wounds of the mice at the same time each day, and the mice were euthanized on the ninth day. The mice were subsequently photographed for skin herpes (Figure 7A), and the width of the skin herpes was counted (Figure 7B). The results showed that both 0.05% AXX-18 gel and 0.1% AXX-18 gel were able to significantly inhibit the width of skin herpes compared with the blank gel group (HSV-1), and there was almost no difference between the 0.1% AXX-18 gel group and the ACV group, suggesting that AXX-18 was able to inhibit the spread of HSV-1-induced skin herpes. To better investigate the anti-HSV-1 activity of AXX-18 in mice, we also examined the changes in viral genes and proteins. Western blot results showed that AXX-18 significantly inhibited the expression of viral protein gD in mouse skin, spleen, and kidney tissues (Figure 7C), indicating that AXX-18 was able to reduce viral load in mouse skin, spleen, and kidney tissues. In addition, qRT-PCR results showed that AXX-18 was able to significantly inhibit the expression of mRNAs of the helicase-primer enzyme genes UL5, UL8, and UL52 (Figure 7D–F), indicating that AXX-18 was still able to inhibit HSV-1 replication in vivo, which is consistent with our previous mechanistic studies. In conclusion, AXX-18 showed good anti-HSV-1 activity both in vivo and ex vivo, and the potential target protein of AXX-18 was clarified, which provides a good scientific basis for our subsequent drug development.

## 4. Discussion

Infection with the herpes simplex virus is very common worldwide. Herpes virus is related to Alzheimer’s disease, and induced neurological encephalitis, tumors, and other diseases [26]. At present, the most widely used therapeutic drug is still ACV, but due to the emergence of drug-resistant strains because of the wide application of ACV, exploring new antiviral drugs is urgent. Currently, monomeric compounds derived from natural products are of high research value because of their low toxicity and multi-targeting characteristics. In this study, the oleanolic-acid derivative, AXX-18, is a monomeric compound derived from benzoin, and studies have reported that oleanolic acid has anti-oxidant, anti-tumor, anti-inflammatory, anti-diabetic, anti-microbial, and anti-viral effects [27,28,29,30,31,32]. Our previous study found that oleanolic acid has good anti-HSV-1 activity and elucidated its antiviral mechanism in detail. In this study, we compared the antiviral activity of AXX-18 and oleanolic acid, exploring the antiviral mechanism of AXX-18.

In order to determine the non-toxic concentration of AXX-18, we conducted a CCK8 assay and found that the CC_50_ values of AXX-18 were higher than OC on both Vero and HaCaT cells, indicating that the toxicity of AXX-18 is lower than OC. In addition, the plaque-reduction assay showed that AXX-18 could achieve 100% plaque-reduction inhibition at 5 μM, whereas OC could only achieve 100% null-spot inhibition at 20 μM, indicating that the antiviral activity of AXX-18 was significantly higher than that of OC. HaCaT cells are human immortalized keratinocytes, a type of human epidermal cell, and the epidermis is the first line of defense when the body is exposed to a virus, so we have carried out subsequent mechanistic studies on HaCaT cells. We first compared the antiviral activity of AXX-18 and oleanolic acid on HaCaT cells, where we examined the copy number of viral genomic DNA, viral titer, and the amount of viral protein expressed, comparing the antiviral activity of AXX-18 with that of OC in a comprehensive manner from three aspects and fully demonstrating that the antiviral activity of AXX-18 was significantly superior to that of OC. Our previous study revealed that OC has activity against ACV-resistant strains [19], so we used the same approach to compare the activity of AXX-18 with OC against ACV-resistant strains on Vero cells. The results showed that the EC_50_ of AXX-18 against HSV-1/106, HSV-1/153, and HSV-1/Blue was 6.04 μM, 9.29 μM, and 6.78 μM respectively, which were lower than that of OC at 12.74 μM, 15.21 μM, and 12.96 μM, suggesting that the activity of AXX-18 against ACV-resistant strains was also significantly stronger than that of OC. In order to fully substantiate this point, we also probed on HaCaT cells, and the results of viral genome-copy number and viral titer showed that the activity of AXX-18 against the ACV-resistant strain was stronger than that of OC. In particular, we measured the expression of both the immediate-early gene α0 and the late gene UL47 when measuring the viral genomic DNA copy number, thus making the results of the viral genomic-DNA-copy-number testing more convincing. Consistent with the study by Paromita Baga et al. [33], the time point at which AXX-18 acts was explored by adding the drug at different time points in the plaque-reduction assay. Our results showed that AXX-18 showed significant plaque inhibition during the 0–24 h period, and there was a significant difference between 2 h and 4 h, while 4 h was almost unchanged compared to 6 h; so, we focused on 2 h and 4 h, which theoretically belong to the immediate-early stages of HSV-1 infection. On the other hand, the research value of a drug would be increased if it were more effective at an earlier stage. The statistics on the size of the empty spots suggest that AXX-18 may have inhibited the cell-to-cell transmission of HSV-1, and we did not investigate this finding in-depth in this study. In addition, we also examined viral genome-copy number and viral titer, and the results showed that AXX-18 exerted higher antiviral activity at an immediately early stage, which is consistent with the time point of OC against HSV-1. Immediately following this, we carried out studies at the immediate-early stage and, unfortunately, AXX-18 did not affect immediate-early gene and protein expression at the immediate-early stage, nor did it affect HSV-1 attachment, penetration, or nuclear entry. This leads us back to the ability of AXX-18 to inhibit viral genomic-DNA replication as well as reduce viral titers; then, we speculate that AXX-18 exerts its antiviral activity by inhibiting viral replication. In addition, AXX-18 significantly inhibited the activity of ACV-resistant strains, suggesting that the antiviral mechanism of AXX-18 is different from that of ACV, so we targeted the helicase-primer enzyme complex (UL5, UL8, UL52). Previously, it has been reported that the helicase-primer enzyme complex could be a potential target for next-generation drugs [34]. Fortunately, AXX-18 was able to significantly suppress the mRNA expression levels of UL5, UL8, and UL52 in the immediate-early stages. This result was similar to that of oleanolic acid, except that AXX-18 could also significantly inhibit the expression of UL5, UL8, and UL52 at 2 h, whereas oleanolic acid could only inhibit the expression of UL5, UL8, and UL52 at 4 h. Subsequently, the potential targets of AXX-18 action were explored by siRNA knock down and overexpression. Surprisingly, AXX-18 did not significantly reduce viral titers when UL8 or UL52 were knocked down. We, therefore, speculated that AXX-18 might act on both UL8 and UL52, and this was confirmed by overexpression reversion experiments, where overexpression of either UL8 or UL52 significantly reverted to the reduction in viral titer caused by AXX-18. AXX-18 significantly suppressed the expression of gD in the Vector group as well as in overexpression of UL5, while the expression of gD was significantly restored by overexpression of UL8 and UL52. It is suggested that UL8 and UL52 may be potential targets of action of AXX-18, whereas the target protein of OC is UL8, excluding UL52 [19], and, therefore, we speculate that this may be the reason why the anti-HSV-1 activity of AXX-18 is superior to that of OC. In addition, we found that there was no significant increase in viral titer when UL5, UL8, and UL52 were overexpressed separately, but there was a significant decrease in viral titer when UL8 or UL52 were knocked down alone, so we speculate that UL5, UL8, and UL52 need to form a complex to significantly promote viral replication, as overexpression of UL8 or UL52 alone does not promote viral replication. However, since UL8 and UL52 are essential for viral replication, knocking down UL8 or UL52 alone significantly reduced viral titers, which is consistent with previous studies.

To better investigate the anti-HSV-1 activity of AXX-18, we constructed a model of skin herpes caused by HSV-1, and after 9 days of treatment with different concentrations of AXX-18, we found a significant reduction in the width of the skin herpes, which was similar to that of ACV. We also examined the viral load in skin, spleen, and kidney tissues; then, we detected the expression of the viral protein gD, which represented the viral load in different tissues, and results showed that both 0.1% AXX-18 and 0.05% AXX-18 significantly reduced the expression of the viral protein gD, suggesting that AXX-18 also has significant anti-HSV-1 activity in mice. Finally, qRT-PCR results showed that AXX-18 could significantly inhibit the expression of HSV-1 helicase-primer enzyme-related genes in skin tissues, which is consistent with the results we obtained at the cellular level. Overall, AXX-18 showed stronger anti-HSV-1 activity both in vitro and in vivo, which was higher than that of oleanolic acid. In addition, the clear target of action of AXX-18 provides a good theoretical basis for our subsequent pharmacological studies.

## Figures and Tables

**Figure 1 viruses-14-01287-f001:**
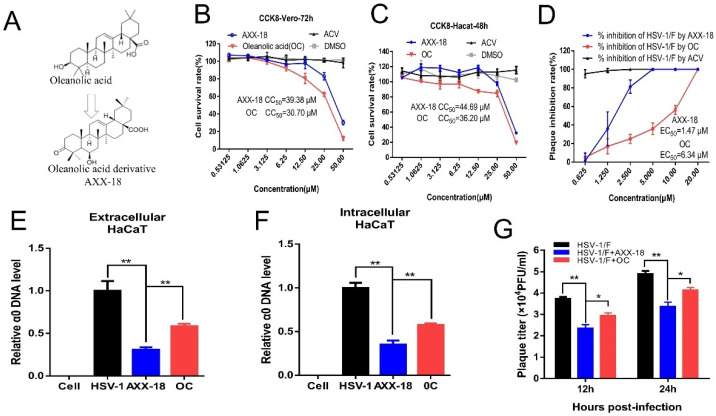
The anti-HSV-1 activity of oleanolic acid derivative AXX-18 is significantly higher than that of oleanolic acid. (**A**) The structure formula of oleanolic acid and oleanolic-acid derivative AXX-18. (**B**,**C**) Vero and HaCaT cells were treated with different concentrations of AXX-18, OC, or ACV for 72 h or 48 h, and cell viability was calculated by CCK8. (**D**) The anti-HSV-1 activity of AXX-18, OC, and ACV on Vero cells. HSV-1 (100 PFU/well) and different concentrations of AXX-18, OC, and ACV were used to treat Vero cells for 72 h and then calculate the plaque suppression rate. (**E**,**F**) HSV-1 (MOI = 1) and AXX-18 (20 μM) or OC (20 μM) were used to treat HaCaT cells for 24 h, extract extracellular (supernatant) and intracellular viral genomic DNA, and detect the copy number of viral genomic DNA α0 by qRT-PCR. (**G**) HSV-1 (MOI = 1) and AXX-18 (20 μM) or OC (20 μM) were used to treat HaCaT cells for 12 h and 24 h, then detect the virus titer by the CPE method. Data are mean ± SD (*n* = 3). * *p* < 0.05, ** *p* < 0.01 versus HSV-1-treated group.

**Figure 2 viruses-14-01287-f002:**
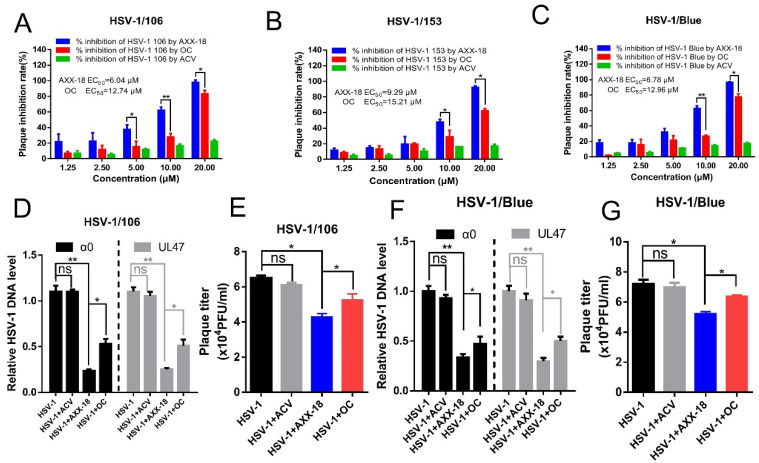
Against ACV-resistant virus strains’ activity of AXX-18, OC, and ACV. (**A**–**C**) Vero cells were infected with HSV-1/106, HSV-1/153, and HSV-1/Blue (100 PFU/well) in the presence of AXX-18, OC, or ACV for 72 h. The plaque numbers were counted to calculate the inhibitory effect. (**D**,**F**) HSV-1/106, HSV-1/Blue (MOI = 1), and AXX-18 (20 μM), OC (20 μM), or ACV (20 μM) were used to treat HaCaT cells for 24 h, then the sample was frozen and thawed three times and viral genomic DNA was extracted, finally the copy number of genomic DNA α0 and UL47 was detected by qRT-PCR. (**E**,**G**) The same treatment as (**D**,**F**) for the preceding and repeated freeze–thaw was used for virus-titer determination by CPE. Data are mean ± SD (*n* = 3). * *p* < 0.05, ** *p* < 0.01, versus HSV-1-treated group.

**Figure 3 viruses-14-01287-f003:**
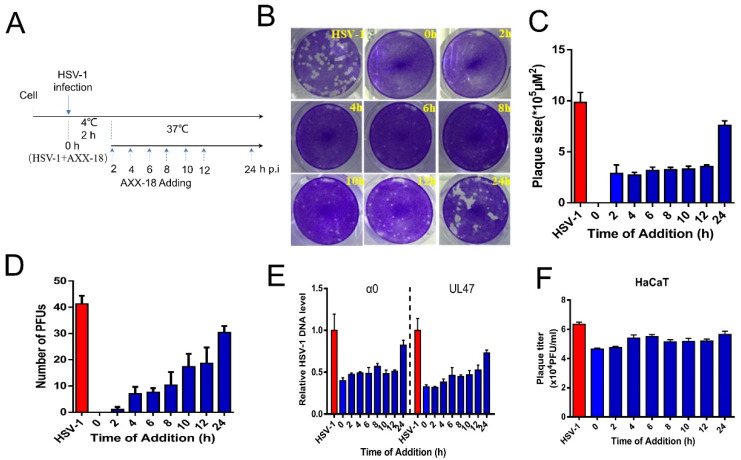
AXX-18 acts in the immediate-early stage. (**A**) The simple diagram of the time-addition assay (**B**) Vero cells were treated with HSV-1 (100 PFU/well) and AXX-18 (20 μM) at 4 °C for 2 h, the covering solution containing the AXX-18 was added at different time points and incubated at 37 °C for 72 h, then the cells were fixed and stained with crystal-violet dye, and the plaque numbers and size of plaques were counted to calculate the inhibition rate. (**C**) Statistics of plaque size by ImageJ. (**D**) Statistics of the number of plaques by ImageJ. (**E**) HaCaT cells were treated with HSV-1 (MOI = 1) and AXX-18 (20 μM) at 4 °C for 2 h, then the AXX-18 was added at the time point, and incubated at 37 °C for 48 h, then freezing and thawing the samples was repeated three times to extract viral genomic DNA, and the copy number of genomic DNA α0 and UL47 was detected by qRT-PCR. (**F**) The previous processing method is the same as (**E**), and the repeated freeze–thaw samples were tested for virus titers by CPE. Data are mean ± SD (*n* = 3).

**Figure 4 viruses-14-01287-f004:**
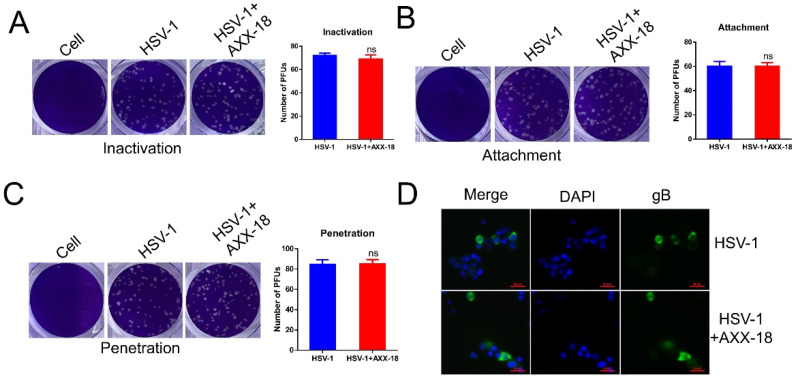
AXX-18 does not affect the attachment, penetration, and nuclear entry of HSV-1. (**A**) HSV-1 (100 PFU/well) and AXX-18 (20 μM) were incubated at 37 °C for 2 h and then infected with HaCaT cells for 2 h. After that, the culture was removed and replaced with a covering solution, and the incubation continued for 72 h. Finally, crystalline violet staining was performed and the number of empty spots was counted. (**B**) HaCaT cells were infected with HSV-1 (100 PFU/well) and AXX-18 mixed solution at 4 °C for 2 h, then HSV-1 was absorbed, discarded, and washed with PBS (pH = 11) and PBS (pH = 3) for 1 min, respectively; then, it was replaced with a covering solution and continued to incubate for 72 h; finally, crystalline violet staining was performed and the number of empty spots was counted. (**C**) HaCaT cells were infected with HSV-1 (100 PFU/well) and AXX-18 mixed solution at 4 °C for 2 h, and then cultured at 37 °C for 10 min. After that, HSV-1 was absorbed and discarded, washed twice with PBS, and cultured for another 72 h with a covering solution. Finally, crystalline violet staining was performed and the number of empty spots was counted. (**D**) HaCaT cells were infected with HSV-1 (MOI = 1) and AXX-18 mixed solution for 2 h at 37 °C, then HSV-1 was absorbed and discarded, washed with PBS three times, and then closed, permeated, and incubated with gB (green) antibody, fluorescent secondary antibody, and DAPI (blue); finally, it was photographed by a fluorescence laser confocal microscope. Data are mean ± SD (*n* = 3).

**Figure 5 viruses-14-01287-f005:**
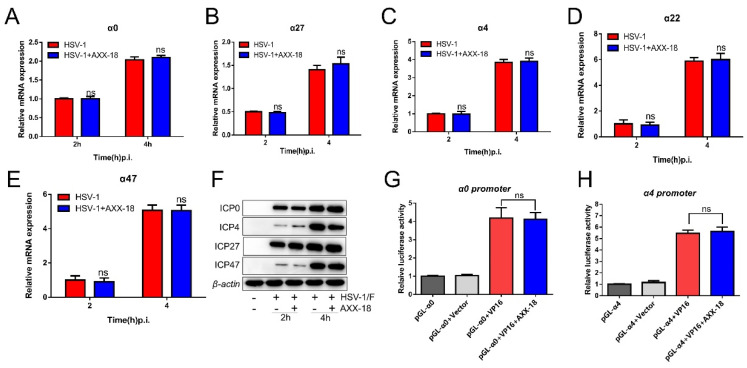
AXX-18 does not affect the immediate-early gene and protein expression of HSV-1. (**A**−**E**) HaCaT cells were treated with HSV-1 (MOI = 5), and AXX-18 (20 μM), and RNA was collected at 2 h and 4 h, respectively. Then, RNA was isolated and subjected to cDNA synthesis, followed by the quantitative qRT-PCR of immediate-early genes using TB Green. (**F**) HaCaT cells were treated with HSV-1 (MOI = 5) and AXX-18 (20 μM), and proteins were collected at 2 h and 4 h, respectively, after extracting the protein and detecting the expression level of immediate-early protein ICP0, ICP4, ICP27, and ICP47 by Western Blot assay. Data are mean ± SD (*n* = 3). (**G**,**H**) HaCaT cells only transfected with Pgl-α0 plasmids (pGL-α4 plasmids) or co-transfected with pGL-α0 and pcDNA plasmids (pGL-α4 and pcDNA plasmids) were treated as the negative control. Data are mean ± SD (*n* = 3).

**Figure 6 viruses-14-01287-f006:**
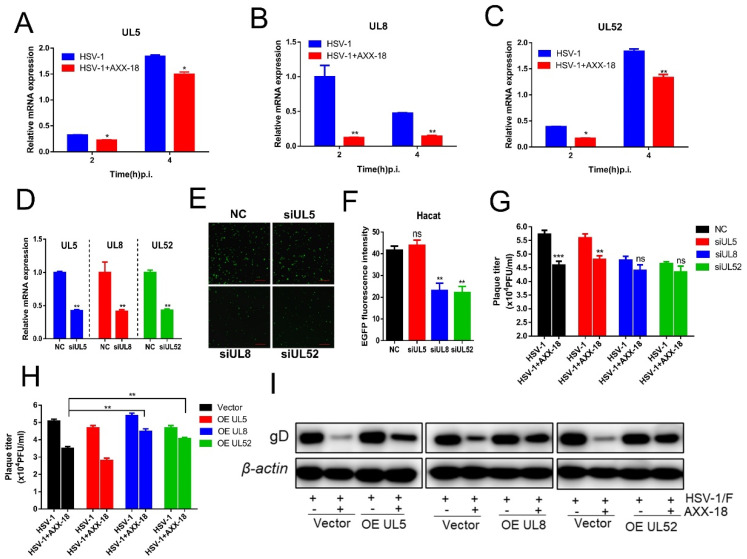
UL8 and UL52 are potential targets for AXX-18 to exert antiviral effect. (**A**–**C**) HaCaT cells were treated with HSV-1 (MOI = 5) and AXX-18 (20μM), and RNA was collected at 2 h and 4 h, respectively, then RNA was isolated and subjected to cDNA synthesis, followed by the quantitative qRT-PCR of UL5, UL8, and UL52 using TB Green. (**D**) HSV−1-infected (MOI = 1) TaCaT cells were treated for 4 h, the siRNAs were transfected into HaCaT separately and continued to be incubated at 37 °C for 24 h, and then RNA was isolated and subjected to cDNA synthesis, followed by the quantitative qRT-PCR of UL5, UL8, and UL52 using TB Green. Data are mean ± SD (*n* = 3). (**E**) HSV-1/EGFP (MOI = 1) infected HaCaT cells for 4 h, then the three siRNAs were transfected into HaCaT separately, continued to be incubated at 37 °C for 24 h, and finally photographed with a fluorescent microscope. (**F**) Statistics of (**E**) fluorescence intensity. (**G**) HSV-1-infected cells were the same as (**D**), and then siRNA was transfected into the cells separately. After 4 h, the contents were aspirated, AXX-18 was added, and incubation was continued for 24 h. Finally, the sample was frozen and thawed three times and the virus titer was checked by CPE. (**H**) UL5, UL8, and UL52 overexpression plasmids were transfected into HaCaT cells for 24 h, followed by infection with HSV-1 (MOI = 1) for 24 h. Finally, the sample was frozen and thawed three times and the virus titer was checked. Data are mean ± SD (*n* = 3). * *p* < 0.05, ** *p* < 0.01, *** *p* < 0.001 versus HSV-1-treated group. (**I**) The previous treatment procedure was the same as (**H**), the proteins were collected after 24 h of AXX-18 treatment, and the expression of the viral proteins gD was measured by Western blot.

**Figure 7 viruses-14-01287-f007:**
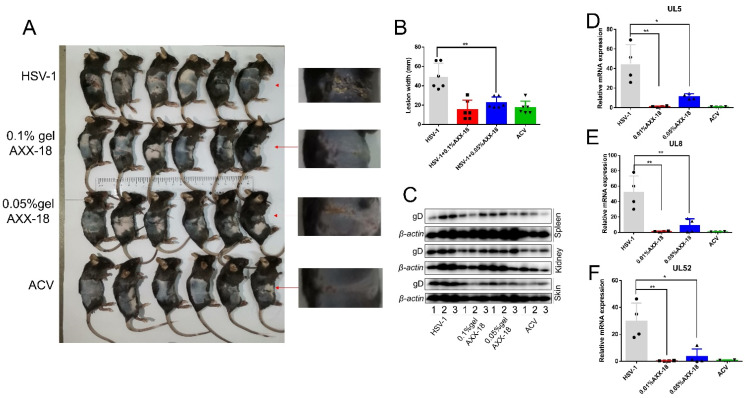
AXX-18 can significantly improve skin herpes caused by HSV-1 infection. (**A**) HSV-1-infected mice were treated with gels containing the indicated concentration of AXX-18 on the infected skin once per day. Mice were euthanized on day 9 after HSV-1 infection, then the zosteriform lesions were observed and photographed, and the widths of the skin herpes were quantified (**B**). (**C**) Separately skin, spleen, and kidney tissue proteins were extracted, and the expression of viral protein gD was detected by Western blot (*n* = 4). (**D**–**F**) The skin of the mouse was taken, RNA was extracted and converted into cDNA, and the mRNA-expression level of UL5, UL8, and UL52 was detected by qRT-PCR (*n* = 4). * *p* < 0.05, ** *p* < 0.01.

## Data Availability

Not applicable.

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
