# Peer review of "Oleanolic Acid Derivative AXX-18 Exerts Antiviral Activity by Inhibiting the Expression of HSV-1 Viral Genes UL8 and UL52"

_viruses, 2022, doi:10.3390/v14061287_

Round 1

Reviewer 1 Report

This study describes experiments evaluating the antiviral activity of the oleanolic acid derivative AXX-18 against herpes simplex virus infection.  Both cell culture and a mouse model of HSV zosteriform spread are utilized. The figures present data showing the antiviral effects of the compound on several aspects of the HSV life cycle.  Overall, the results show that AXX-18 can reduce virus replication at an early step in the life cycle and that it can impede the replication of acyclovir resistant HSV variants.  There are several areas for improvement needed in the text of the manuscript, which are listed below.

1.     The authors conclude that AXX-18 significantly inhibited herpes zoster, however, herpes zoster was not tested in this study.  Therefore, references to herpes zoster and shingles need to be removed from the manuscript.  Herpes zoster, also known as Shingles, is caused by a reactivation of the varicella zoster virus.  Only HSV was tested in the experiments presented.  Specifically, the authors tested the antiviral activity of HSV in a murine zosteriform spread model.  All mentions of “herpes zoster” should, therefore, be removed and replaced with something like “HSV zosteriform spread”.

2.     What is the gel used for the mouse experiments?  The constituents of the gel to which the AXX-18 was added need to be included in the materials and methods section.  Was there a group of mice treated with HSV and gel alone (no AXX-18)?  If not, what is the evidence that the reduction in viral replication is due to the presence of AXX-18 and not the gel alone?

3.     In the description of Figure 3, the authors indicate that AXX-18 is most efficient when added between 0-4 hpi, however, the data presented appears to show little difference between the addition of drug at 4 and 6 hours for panel D – E, and between 4 – 12 hours in panels C and F. This discrepancy should be addressed.

4.     The experimental design for the experiments in Figure 5 -6 was unclear to me based on the descriptions in the Figure legends.  Specifically, it indicates that “infected cells were treated with AXX-18 for 2 and 4 h post-infection.”  Does that mean that the infection was allowed to proceed for two hours and then the AXX-18 was added, or was the virus and drug added together?  How long after the infection did the harvest take place?  A diagram similar to that shown in Figure 3 may be helpful.

5.     Have the ACV resistant mutants used here been previously characterized?  Do all mutations occur within the TK gene? References to the mutation characterization should be cited, if available. 

6.      In line 40, the authors list AIDS as a disease related to HSV.  Without additional context, this is misleading as it implies that AIDS is caused by HSV.  Please add additional information to describe the relationship between AIDS and HSV or remove AIDS from the list of disease here.

7.     Line 505 refers to “HSV-1 decapentaplegic-initiation-enzyme related genes”.  What is this?  It is not described elsewhere in the manuscript.

8.     On lines 60 and 293, the phrase “We all know that” should be removed.  Also “It is we all known that” on line 324 should be removed.

9.     The manuscript should be checked throughout for grammatical and spelling errors.  Some examples are listed below.

      a.     In several places within the text, the word “nucleus” or “nuclear” is used incorrectly as a verb.  For example, in line 44 the term “nucleus” should be replaced with “nuclear entry”.  This should be corrected throughout the manuscript.

b.     There are several instances of the authors saying that the antiviral activity of     AXX-18 is “better” than OC. The term “better” should be replaced with a more precise term, such as “ higher” or “stronger”.

c. Figure 1G, “hours” is spelled wrong.

Author Response

Response to Reviewer 1 Comments

This study describes experiments evaluating the antiviral activity of the oleanolic acid derivative AXX-18 against herpes simplex virus infection.  Both cell culture and a mouse model of HSV zosteriform spread are utilized. The figures present data showing the antiviral effects of the compound on several aspects of the HSV life cycle.  Overall, the results show that AXX-18 can reduce virus replication at an early step in the life cycle and that it can impede the replication of acyclovir resistant HSV variants.  There are several areas for improvement needed in the text of the manuscript, which are listed below.

  1. The authors conclude that AXX-18 significantly inhibited herpes zoster, however, herpes zoster was not tested in this study.  Therefore, references to herpes zoster and shingles need to be removed from the manuscript.  Herpes zoster, also known as Shingles, is caused by a reactivation of the varicella zoster virus.  Only HSV was tested in the experiments presented.  Specifically, the authors tested the antiviral activity of HSV in a murine zosteriform spread model.  All mentions of “herpes zoster” should, therefore, be removed and replaced with something like “HSV zosteriform spread”.

Response 1: Based on your suggestion, we have a deep understanding of " herpes zoster ", which is indeed caused by the chickenpox virus, so our description here is inaccurate and we have uniformly replaced shingles with skin herpes caused by HSV-1 in the manuscript. Thanks again for your suggestion.

  1. What is the gel used for the mouse experiments?  The constituents of the gel to which the AXX-18 was added need to be included in the materials and methods section.  Was there a group of mice treated with HSV and gel alone (no AXX-18)?  If not, what is the evidence that the reduction in viral replication is due to the presence of AXX-18 and not the gel alone?

Response 2: The gel composition of AXX-18 gel formulation is composed of glycerol (G10007) 99% and carbomer (M64011) 1%. The safety of carbomer has been verified in many aspects. Relevant literature is cited in the manuscript to demonstrate the safety of carbomer. In fact, the HSV-1 group was treated with a blank gel, and compared with the administration group of 0.1% AXX-18, 0.05% AXX-18 and ACV, it can be clearly seen that different concentrations of AXX-18 can significantly improve the skin herpes in mice, this suggests that the treatment of skin herpes in mice is the effect of AXX-18.

Thanks for your suggestion, we have supplemented the manuscript with a description of the HSV-1 group.

  1. In the description of Figure 3, the authors indicate that AXX-18 is most efficient when added between 0-4 hpi, however, the data presented appears to show little difference between the addition of drug at 4 and 6 hours for panel D – E, and between 4 – 12 hours in panels C and F. This discrepancy should be addressed.

Response 3: As you mentioned, there was no significant difference in viral titer and genome copy number between the 4h and 6h groups (but still showed good antiviral activity compared to the control group), while the 0h and 2h groups were least significant across the time axis (0-24h). There was a significant difference between the 2h and 4h groups, which was also the focus of our attention, so we chose these two time points for the study, while there was no difference between the 4h and 6h time points was not the focus of our attention. Maybe our description in the result was not clear enough, so we made a supplement to this point in the discussion. Thank you very much for your suggestion.

  1. The experimental design for the experiments in Figure 5 -6 was unclear to me based on the descriptions in the Figure legends.  Specifically, it indicates that “infected cells were treated with AXX-18 for 2 and 4 h post-infection.”  Does that mean that the infection was allowed to proceed for two hours and then the AXX-18 was added, or was the virus and drug added together?  How long after the infection did the harvest take place?  A diagram similar to that shown in Figure 3 may be helpful.

Response 4: For the experimental design of Figure5-6, we uniformly infected cells with AXX-18 and HSV-1, and collected RNA and protein 2h and 4h later, respectively. Maybe our description in the annotation is not clear enough, so we have modified it in the manuscript.

  1. Have the ACV resistant mutants used here been previously characterized?  Do all mutations occur within the TK gene? References to the mutation characterization should be cited, if available. 

Response 5: Thank you for your question. The other two ACV-resistant strains also have TK kinase mutations, which we have modified and correctly quoted relevant literature in the manuscript.

  1. In line 40, the authors list AIDS as a disease related to HSV.  Without additional context, this is misleading as it implies that AIDS is caused by HSV.  Please add additional information to describe the relationship between AIDS and HSV or remove AIDS from the list of disease here.

Response 6: You are right. What we have described here is not appropriate enough and it is misleading. According to your suggestion, we deleted AIDS from the manuscript. thank you.

  1. Line 505 refers to “HSV-1 decapentaplegic-initiation-enzyme related genes”.  What is this?  It is not described elsewhere in the manuscript.

Response 7: This is a mistake in our writing and we have corrected it in the manuscript. Thank you very much.

  1. On lines 60 and 293, the phrase “We all know that” should be removed.  Also “It is we all known that” on line 324 should be removed.

Response 8: Thank you for your suggestion. We have deleted them from the manuscript.

  1. The manuscript should be checked throughout for grammatical and spelling errors.  Some examples are listed below. (1) In several places within the text, the word “nucleus” or “nuclear” is used incorrectly as a verb.  For example, in line 44 the term “nucleus” should be replaced with “nuclear entry”.  This should be corrected throughout the manuscript. (2)There are several instances of the authors saying that the antiviral activity of  AXX-18 is “better” than OC. The term “better” should be replaced with a more precise term, such as “ higher” or “stronger”. (3) Figure 1G, “hours” is spelled wrong.

Response 9: We have carefully reviewed the manuscript and corrected the grammar errors, which were indicated in the manuscript. Thank you for your advice.

Please see the attachment (Manuscript)

Reviewer 2 Report

The manuscript “Oleanolic acid derivative AXX-18 exerts antiviral activity by inhibiting the expression of viral genes UL8 and UL52” by Zhaoyang Wang, et al., is interesting and potentially valuable for therapy of herpes viruses. I think that the authors should clarify some points:

  1. Some terms should be clarified in the abstract (strongly L17; many L18; formation of herpes zoster L27; ACV and so on L19).
  2. In the title I think that the term HS1 is missing (perhaps…. HS1 viral genes…).
  3. Contrary to the idea that this work “clarified its mechanism of action” (L29) it showed the anti-viral activity of AXX-18.
  4. I think that the “in vivo” experiments (L387-394) should be first explained in the section of methods and then the results in the corresponding site (L395-408).
  5. At the end of Discussion, it would be important to point out that pharmacological studies could be conducted in order to determine the potential use of AXX-18 in the therapy of herpes infections.

Author Response

Response to Reviewer 2 Comments

  1. Some terms should be clarified in the abstract (strongly L17; many L18; formation of herpes zoster L27; ACV and so on L19).

Response 1: The question you raised is very important, and we have added and revised it in the manuscript, thank you.

  1. In the title I think that the term HS1 is missing (perhaps…. HS1 viral genes…).

Response 2: You are right, we have changed the title to “Oleanolic acid derivative AXX-18 exerts antiviral activity by inhibiting the expression of HSV-1 viral genes UL8 and UL52”. Thank you.

  1. Contrary to the idea that this work “clarified its mechanism of action” (L29) it showed the anti-viral activity of AXX-18.

Response 3: After your prompting we found that the description here may not be accurate, so we have replaced " have clarified its mechanism of action " with " identification of its potential target proteins ". This may be a more accurate description. Thank you for your suggestion.

  1. I think that the “in vivo” experiments (L387-394) should be first explained in the section of methods and then the results in the corresponding site (L395-408).

Response 4: We show the specific experimental process in the legend, and modify the experimental results part, thank you for your suggestion.

  1. At the end of Discussion, it would be important to point out that pharmacological studies could be conducted in order to determine the potential use of AXX-18 in the therapy of herpes infections.

Response 5: Thank you for your suggestions, we have made changes in the manuscript.

Please see the attachment (Manuscript)

Round 2

Reviewer 1 Report

The authors have addressed most of the questions/concerns from the first review. Comments on remaining issues are provided below.

1.         There is still one reference to shingles in the manuscript.  This is in the legend to Figure 7.  “F)Take the skin of mouse shingles…”  This should be removed.

2.           Please define “NC” for Figure 6.

3.           It is unclear what is meant by “snapped” in the legend to Figure 7.   Do you mean photographed, captured an image, or something else?

4.           In Figure 6I, there appears to be a label missing to indicate what the  “+” and “– “ signs refer to.  Also, in the legend to Figure 6, I believe “TaCaT” should be “HaCaT”.

5.           There are still several instances of grammatical errors in the manuscript.  Examples are provided below.  It is recommended that careful proofreading of the entire manuscript is done to correct these types of errors e.g., inappropriate word choice and verb tense.

a.           On page 9, the bolded header for figure 4 should say “nuclear entry” instead of just “nuclear”

b.           In the legend for Figure 7, “invert” should be replaced with “convert” or “reverse transcribe”.

c.           In the first paragraph of the discussion section, the following sentence contains grammatical errors and should be revised. “Current obtained from natural products have become the research hotspot monomeric compounds because such monomer compounds generally have low toxicity, and multiple target characteristics, the present study drug is oleanolic acid derivative (AXX-18), extracted from benzoin.”

d.           In figure labels “Relative of HSV-1 DNA level” should be replaced with “Relative HSV-1 DNA level.”

Author Response

The authors have addressed most of the questions/concerns from the first review.

Comments on remaining issues are provided below.

  1. There is still one reference to shingles in the manuscript.  This is in the legend to Figure 7.  “F) Take the skin of mouse shingles…”  This should be removed.

Response 1: Thanks for your review of our manuscript, we have removed the shingles.

  1. Please define “NC” for Figure 6.

Response 2: Thanks for your suggestion, we have defined NC in the manuscript and shown in red.

  1. It is unclear what is meant by “snapped” in the legend to Figure 7.   Do you mean photographed, captured an image, or something else?

Response 3: Yes, "snapped" in this case means to take a picture of a mouse, for clarity we have replaced "snapped" with "photographed".

  1. In Figure 6I, there appears to be a label missing to indicate what the  “+” and “– “ signs refer to.  Also, in the legend to Figure 6, I believe “TaCaT” should be “HaCaT”.

Response 4: Thanks for your suggestion, we have corrected it in the manuscript.

  1. There are still several instances of grammatical errors in the manuscript.  Examples are provided below.  It is recommended that careful proofreading of the entire manuscript is done to correct these types of errors e.g., inappropriate word choice and verb tense.
  2. On page 9, the bolded header for figure 4 should say “nuclear entry” instead of just “nuclear”
  3. In the legend for Figure 7, “invert” should be replaced with “convert” or “reverse transcribe”.
  4. In the first paragraph of the discussion section, the following sentence contains grammatical errors and should be revised. “Current obtained from natural products have become the research hotspot monomeric compounds because such monomer compounds generally have low toxicity, and multiple target characteristics, the present study drug is oleanolic acid derivative (AXX-18), extracted from benzoin.”
  5. In figure labels “Relative of HSV-1 DNA level” should be replaced with “Relative HSV-1 DNA level.”

Response 5: Thanks for your suggestion, we have revised the manuscript and double checked the grammar, thanks again
